# Optimizing Language Models for Human Preferences is a Causal Inference Problem

Victoria Lin[1]     Eli Ben-Michael[1]     Louis-Philippe Morency[1]

[1]Carnegie Mellon University

## Abstract

As large language models (LLMs) see greater use in academic and commercial settings, there is increasing interest in methods that allow language models to generate texts aligned with human preferences. In this paper, we present an initial exploration of language model optimization for human preferences from *direct outcome datasets*, where each sample consists of a text and an associated numerical outcome measuring the reader's response. We first propose that language model optimization should be viewed as a *causal problem* to ensure that the model correctly learns the relationship between the text and the outcome. We formalize this causal language optimization problem, and we develop a method—*causal preference optimization* (CPO)—that solves an unbiased surrogate objective for the problem. We further extend CPO with *doubly robust* CPO (DR-CPO), which reduces the variance of the surrogate objective while retaining provably strong guarantees on bias. Finally, we empirically demonstrate the effectiveness of (DR-)CPO in optimizing state-of-the-art LLMs for human preferences on direct outcome data, and we validate the robustness of DR-CPO under difficult confounding conditions.

## 1 INTRODUCTION

Recent advances in computation have yielded large-scale self-supervised language models that achieve impressive performance on a variety of natural language processing (NLP) tasks [Zhang et al., 2022, Chowdhery et al., 2023, Scao et al., 2023, Bubeck et al., 2023]. These large language models (LLMs)—trained on vast amounts of text data of varying quality—can acquire less desirable attributes from these texts, and so they often require further fine-tuning on human preferences to improve their factual correctness and alignment with social values (e.g., less toxic, more helpful) [Ouyang et al., 2022, Bommasani et al., 2022].

In this paper, we examine a paradigm for language model optimization for human preferences that has previously been underexplored: learning from *direct outcome datasets*, which are ubiquitous in NLP. In contrast to paired completion data consisting of prompts followed by one preferred and one non-preferred completion, direct outcome datasets are text datasets where each sample consists of a text and an associated numerical *outcome* measuring the reader's response to the text (e.g., Reddit upvotes [Lakkaraju et al., 2021], Amazon ratings [McAuley and Leskovec, 2013]). A large number of direct outcome datasets are *crowdsourced datasets*, where annotators on a crowdsourcing platform are randomly assigned to read and respond to texts.

The ability to learn human preferences from direct outcome data significantly broadens the scope of problems that can be addressed by learning from human feedback. Consider the task of inducing a language model to unlearn hate speech. Current unlearning approaches typically use paired data in the format *([hate speech], [alternative text])*, with the preferred text being the latter. Constructing alternative texts can be difficult [Eldan and Russinovich, 2023, Maini et al., 2024], and rather than fully unlearning the hate speech, the language model is instead trained to preferentially generate the alternative text [Patil et al., 2024]. Direct outcome data, by contrast, allows texts to be directly marked as hateful and removed from the language model without learning or requiring an alternative text. Moreover, direct outcome datasets greatly outnumber paired completion datasets in NLP: at the time of writing, for instance, the HuggingFace Datasets hub contains thousands of direct outcome datasets and fewer than 50 paired completion datasets.

We present an initial exploration of language model optimization in the direct outcome setting, where the language model is fine-tuned to optimize texts with respect to a desired outcome. We first note that learning an optimal lan-

guage model can be difficult due to the presence of un-measured *confounding* in the training data: external factors that affect both readers' choice of texts to read and how they tend to respond to those texts. Language models optimized on confounded data may learn incorrect relationships between texts and reader responses, leading them to generate sub-optimal text. For instance, users of hate speech are both (i) more likely to engage with content containing hate speech and (ii) more likely to rate hateful content positively. Such confounding may lead to incomplete unlearning of hate speech, as some examples of hate speech are assigned positive outcomes in the confounded data.

Therefore, we posit that language model optimization should be viewed as a *causal* problem in order to ensure that the optimal language model *causes* preferred outcomes. In this paper, we introduce a novel causal formulation of the language model optimization problem. The solution to this optimization problem finds how to *intervene* on the text distribution of the generating model to best *cause* an optimal outcome (i.e., generation of human-preferred texts).

We observe that in the direct outcome setting, it is possible in practice to guarantee that the observed relationship between the text and the outcome is causal by leveraging crowdsourced datasets. Due to random assignment of texts to readers, crowdsourced datasets are not subject to external confounding and can in fact be viewed as randomized experiments [Lin et al., 2023]. Building on this observation, we present two methodological contributions that enable causal language model optimization on direct outcome datasets. First, we develop *causal preference optimization* (CPO), which solves an unbiased surrogate objective for the causal optimization problem. Next, we extend this to *doubly robust* CPO (DR-CPO), which improves on CPO by reducing the variance of the surrogate objective via outcome modeling while retaining provably strong guarantees on bias.[1]

We empirically assess the effectiveness of (DR-)CPO in optimizing state-of-the-art LLMs for human preferences on direct outcome data, both with and without confounding. We find that CPO methods successfully optimize LLMs for human preferences and outperform baselines, and we further observe empirical evidence for the robustness of DR-CPO under difficult confounding conditions.

## 2 RELATED WORK

### 2.1 LANGUAGE MODEL OPTIMIZATION

The performance of large self-supervised language models can be further improved by fine-tuning on datasets that align them with human-preferred text [Ouyang et al., 2022, Bommasani et al., 2022]. These *paired completion datasets*

typically consist of prompts followed by two candidate completions, one of which is indicated to be human-preferred [Ethayarajh et al., 2022, Bai et al., 2022, Ji et al., 2023]. A reinforcement learning algorithm may then derive its reward model from these datasets (reinforcement learning from human feedback, or RLHF) [Christiano et al., 2017], after which language models are fine-tuned to maximize the human preference reward under the RLHF algorithm.

While RLHF has seen widespread use [Stiennon et al., 2020, Touvron et al., 2023], it is computationally demanding, as its training loop requires that new texts be generated and new rewards be computed at each step. Consequently, in recent months, methods that allow language models to learn more directly from human preference data have emerged [Hejna et al., 2023, Dumoulin et al., 2024]—the most popular of which is direct preference optimization (DPO) [Rafailov et al., 2023]. Like RLHF, DPO is designed for use with paired completion datasets, maximizing the probability ratio of preferred completions to non-preferred completions over the paired completion dataset.

### 2.2 CAUSAL INFERENCE AND DOUBLY ROBUST POLICY LEARNING

Although RLHF and DPO constitute the two most popular optimization approaches for language models, there exists a wide body of work on estimation and policy learning outside of the NLP space. Some notable work relevant to this paper includes a long history of doubly robust estimation of causal effects [Robins et al., 1994] and—more directly applicably—doubly robust policy learning [Dudík et al., 2011, Jiang and Li, 2016, Tang et al., 2020, Athey and Wager, 2021, Kallus et al., 2022].

In causal inference, double robustness denotes an estimator formulation that provides robustness against misspecification of *nuisance* parameters or functions. In particular, doubly robust estimators combine two existing estimators—an *importance weighting* estimator and an *outcome modeling* estimator—such that only one of the two components must be correctly specified or estimated to guarantee the unbiasedness of the estimator [Robins et al., 1994, Chernozhukov et al., 2022]. This can also be viewed as the importance weighting term providing a *bias correction* for the outcome modeling term. The principle of double robustness can be extended to not only the estimation of causal effects but also the estimation of any quantity, including loss functions or policy objectives, as we do here.

## 3 A CAUSAL VIEW OF LANGUAGE MODEL OPTIMIZATION

When a language model $f$ is trained or fine-tuned to generate texts that are consistent with human preferences, the

---

[1]Our code is publicly available at `https://github.com/torylin/causal-preference-optimization`.

implicit goal can be seen as optimizing texts $X \sim P^f$—texts generated from the model $f$—with respect to some outcome $Y$. In this paper, we consider a direct outcome data format $\mathcal{D}_O = \{(X_1, Y_1), \ldots, (X_n, Y_n)\}$, where $X_i$ is a text that individual $i$ interacts with and $Y_i$ is any numerical response of the individual to the texts (e.g., ratings, either binary or scalar).

An association-based approach to optimize model $f$ is to generate texts that are similar to those that have high outcomes in the dataset, i.e.,

$$\arg\max_f \mathbb{E}_{X \sim P^f}[E_{\mathcal{D}_O}[Y|X]], \qquad (1)$$

where the conditional expectation $\mathbb{E}_{\mathcal{D}_O}[Y|X]$ is the average outcome among individuals who observed the text $X$ and can be learned from $\mathcal{D}_0$. A language model optimized under Equation (1) will generate texts that are *correlated* with high outcomes. We distinguish this from our true optimization goal: to learn—across all possible texts and outcomes—to *intervene* on the distribution of the generating language model to *cause* the best possible outcomes.

These hypothetical outcomes can be formalized using the potential outcomes framework [Neyman, 1923 [1990], Rubin, 1974]: over text space $\mathcal{X}$, for each individual $i$ we posit the existence of a *potential outcome function* $Y_i : \mathcal{X} \to \mathbb{R}$, where $Y_i(x)$ encodes their potential real-valued response if given text $x$.[2] We emphasize that most individuals' potential outcomes are not observed, and so $Y_i(x)$ denotes the response individual $i$ would have had *had they seen text $x$*, possibly contrary to reality. This is also commonly known as the *counterfactual*.

We assume that we sample individuals from a population $\mathcal{G}$ so that the set of potential outcomes is given by $\{Y(x) \mid x \in \mathcal{X}\} \sim \mathcal{G}$. We define $g(x) \equiv \mathbb{E}_{Y(\cdot) \sim \mathcal{G}}[Y(x)]$ as the average outcome if *all* individuals in the population were given text $x$. Note that $g(x)$ is different from the correlational measure $\mathbb{E}_{\mathcal{D}_O}[Y|X]$ because for $\mathbb{E}_{\mathcal{D}_O}[Y|X]$, the association between $X$ and $Y$ may be confounded by an external factor.

Formally, then, our goal is to find a language model $f$ that causes high outcomes $Y$ on average across the population of individuals $\mathcal{G}$ and across the texts generated according to the model. We encode the quality of a generative text model $f$ via its value function that measures the expected outcome (or reward):

$$V(f) \equiv \mathbb{E}_{X \sim P^f}[\mathbb{E}_{Y(\cdot) \sim \mathcal{G}}[Y(X)]] = \mathbb{E}_{X \sim P^f}[g(X)] \quad (2)$$

Then the *causal* optimization problem is to find the language model $f$ that maximizes the expected outcome if a random

individual were given a random text according to $P^f$:

$$\arg\max_f V(f) = \arg\max_f \mathbb{E}_{X \sim P^f}[\mathbb{E}_{Y(\cdot) \sim \mathcal{G}}[Y(X)]] \quad (3)$$

In intuitive terms, this optimization problem asks the following question: which texts would we generate if we knew what every individual's response to every text would be? By optimizing the text with respect to *every possible response*, this construction removes confounding influences on which texts are read or observed, such that the content of the text must be the sole factor that causes the outcome.

# 4 (DOUBLY ROBUST) CAUSAL PREFERENCE OPTIMIZATION

Reframing our optimization problem as a causal inference problem allows us to draw on solutions from statistical causal inference—in particular, the use of randomized experiments to identify causal effects and approximate observing all potential outcomes.[3] We formalize such a randomized experiment and/or crowdsourced annotated dataset as $\mathcal{D}_R = \{(X_1, Y_1), \ldots, (X_n, Y_n)\}$ where texts $X_i$ are drawn i.i.d. from a known randomization distribution $P^R$, and individuals with potential outcome functions $Y_i(\cdot)$ are drawn i.i.d. from the population $\mathcal{G}$.[4] This induces a distribution on the observed responses $Y_i = Y_i(X_i)$ that we denote as $P_y^R$.

Random assignment of individuals to texts removes all confounding outside of the text, since no external factors influence which texts the individual reads. Formally, we have that the texts are independent of the full set of potential outcomes, i.e., $\{Y(x) \mid x \in \mathcal{X}\} \perp\!\!\!\perp X$. We also require a technical assumption that there is *overlap* between the randomization distribution $P^R$ and the distribution $P^f$ generated by the language model we are optimizing: that is, if $P^R(x) = 0$, then $P^f(X) = 0$ as well. This ensures that the randomization distribution is sufficiently informative about the domain we want to optimize over. In principle, this is directly enforceable as a constraint on the language model. In practice, due to the underlying structure of text data and the fact that we often fine-tune language models to the text domain as a precursor to optimization, the overlap assumption is unlikely to be binding.

In this section, we describe *causal preference optimization* (CPO), which solves an unbiased surrogate objective for the true causal optimization problem using importance weighting. Following this definition, we extend CPO using the principle of double robustness, in which we use outcome

---

[2]This notation implicitly rules out the possibility that an individual's responses can be affected by the texts given to others—a common assumption in causal inference [Rubin, 1974].

---

[3]As discussed above, crowdsourced datasets are randomized experiments, since annotators are randomly assigned to read texts.

[4]The probability of any text $X_i$ is known according to the randomization probability. For instance, if each reader reads one text uniformly assigned from a corpus of size $n$, then $P^R(X_i) = \frac{1}{n}$ for all $X_i$ in the corpus.

modeling to reduce the variance of the CPO objective while retaining strong guarantees on bias.

Derivations and technical results are shown in Appendix A.

## 4.1 CAUSAL PREFERENCE OPTIMIZATION

**Identification.** The value of the language model $V(f)$ is a causal quantity that involves the potential outcomes for all individuals, some of which are unobserved. However, we can link the value function to the randomization dataset $\mathcal{D}_R$ (i.e., *identify* it from the observed data) by writing in the following way.

**Proposition 4.1.** *The value function $V(f)$ can be identified as*

$$V(f) = \mathbb{E}_{X \sim P^R}\left[\mathbb{E}_{Y \sim P_y^R}\left[\frac{P^f(X)}{P^R(X)}Y\right]\right] \qquad (V_{IPW})$$

This value function draws on importance weighting principles from statistical causal inference (also referred to as IPW). Observed outcomes $Y \sim P_y^R$ are weighted by the density ratios between texts drawn from the language model $X \sim P^f$ and texts drawn from the randomization distribution $X \sim P^R$; this approximates the average outcome under $P^f$, which is not observed.

**Estimation.** After writing the causal quantity $V(f)$ in terms of observable data, we focus on estimating $V(f)$ in practice. The importance weighting value function $V_{IPW}(f)$ can be estimated directly from the crowdsourced data $\mathcal{D}_R$ as follows (recall that $X_i \sim P^R, Y_i \sim P_y^R$):

$$\widehat{V}_{IPW}(f) = \frac{1}{n}\sum_{i=1}^{n}\frac{P^f(X_i)}{P^R(X_i)}Y_i$$

Note that both $P^f$ and $P^R$ are *known* quantities and do not need to be estimated—$P^f$ because it is obtained directly from the model $f$ we are optimizing, and $P^R$ because we know the randomization mechanism of the texts in $\mathcal{D}_R$.[5] Importantly, this means that $\widehat{V}_{IPW}(f)$ is an unbiased estimator for $V(f)$.

**Theorem 4.2.** *Let $\mathcal{D}_R$ be a randomized experiment parameterized by $P^R$, such that $P^R$ is known. Then*

$$\mathbb{E}_{Y(\cdot)\sim\mathcal{G}}[\mathbb{E}_X[\widehat{V}_{IPW}(f)]] = \mathbb{E}_{X\sim P^f}[\mathbb{E}_{Y(\cdot)\sim\mathcal{G}}[Y(X)]]$$
$$= V(f)$$

---

[5]In practice, it can still be empirically helpful to use a model-derived estimate of the randomization probabilities $\widehat{P}^R(X)$, similar to how the Hájek estimator can have lower variance than the Horvitz-Thompson estimator [Hájek, 1971, Särndal et al., 2003].

## 4.2 DOUBLY ROBUST CAUSAL PREFERENCE OPTIMIZATION

An importance weighting estimator like the CPO value function is a natural solution for estimating causal quantities. However, CPO optimizes over only randomized experimental data such as crowdsourced data, and so it can be further improved by the addition of an outcome modeling term that can leverage (often larger) non-randomized data to learn to predict outcomes on unlabeled texts. *Doubly robust causal preference optimization* (DR-CPO) combines IPW (over randomized data) and outcome modeling (over non-randomized data) to yield a doubly robust estimator that reduces the variance of CPO and improves its generality while still remaining unbiased for the true causal optimization problem.

**Identification.** The doubly robust formulation gives us another way of linking the value function to the randomization dataset $\mathcal{D}_R$.

**Proposition 4.3.** *The value function $V(f)$ can also be identified as*

$$V(f) = \mathbb{E}_{X\sim P^R}\left[\mathbb{E}_{Y\sim P_y^R}\left[\frac{P^f(X)}{P^R(X)}(Y - g(X))\right]\right] +$$
$$\mathbb{E}_{X\sim P^f}[g(X)] \qquad (V_{DR})$$

This construction combines IPW and an outcome model $g$ to provide robustness against misspecification or misestimation within either term—akin to doubly robust estimators that serve the same purpose when estimating causal effects.

**Estimation.** The doubly robust value function $V_{DR}$ can be estimated from the crowdsourced data $\mathcal{D}_R$ and a learned outcome model $\widehat{g}(X)$.

First, however, we consider the outcome modeling term $\mathbb{E}_{X\sim P^f}[g(X)]$. Even if we were to have access to the true outcome model $g$, it is difficult to optimize $g$ with respect to texts $X \sim P^f$, as this requires that texts be drawn from the language model $f$ *as $f$ is being updated*. To remedy this, we re-write $\mathbb{E}_{X\sim P^f}[g(X)]$ in terms of a fixed language model $f^0$:[6]

$$\mathbb{E}_{X\sim P^f}[g(X)] = \mathbb{E}_{X\sim P^{f^0}}\left[\frac{P^f(X)}{P^{f^0}(X)}g(X)\right] \qquad (V_{out})$$

where $P^{f_0}$ denotes the distribution over texts from language model $f_0$.

We can create a Monte Carlo estimate of this by drawing texts $\widetilde{X}_1, \ldots, \widetilde{X}_m \sim P^{f^0}$ and computing

$$\widehat{V}_{out}(f) = \frac{1}{m}\sum_{j=1}^{m}\frac{P^f(\widetilde{X}_j)}{P^{f^0}(\widetilde{X}_j)}\widehat{g}(\widetilde{X}_j)$$

---

[6]We show this equivalence in Appendix A.3.

where $\widehat{g}(x)$ is a model trained to predict $Y$ from $X$ and $f^0$ is any generative language model.

Finally, the doubly robust value function $V_{DR}$ can be estimated as a combination of these two terms.

$$\widehat{V}_{DR}(f) = \frac{1}{n}\sum_{i=1}^{n}\frac{P^f(X_i)}{P^R(X_i)}(Y_i - \widehat{g}(X_i)) +$$
$$\frac{1}{m}\sum_{j=1}^{m}\frac{P^f(\widetilde{X}_j)}{P^{f^0}(\widetilde{X}_j)}\widehat{g}(\widetilde{X}_j)$$

Formally, it can be shown that $\widehat{V}_{DR}(f)$ is an unbiased estimator for $V(f)$ under two possible conditions, making it an effective proxy for the true causal optimization problem.

**Theorem 4.4.** *Let $\mathcal{D}_R$ be a randomized experiment parameterized by $P^R$, which may be estimated from a separate sample by $\widehat{P}^R$. Let $g(X) = E_{Y(\cdot)\sim\mathcal{G}}[Y(X)]$, which may be estimated from a separate sample by $\widehat{g}(X)$. Then*

$$\mathbb{E}_{Y(\cdot)\sim\mathcal{G}}[\mathbb{E}_X[\widehat{V}_{DR}(f)]] = \mathbb{E}_{X\sim P^f}[\mathbb{E}_{Y(\cdot)\sim\mathcal{G}}[Y(X)]]$$
$$= V(f)$$

*if either*

1. $\widehat{P}^R(X) = P^R(X)$, or
2. $\widehat{g}(X) = g(X)$

Importantly, because $P^R$ *is known* in randomized experiments, including the crowdsourced data setting, it does not need to be estimated, which means that condition (1) is always fulfilled for DR-CPO. As a result, $\widehat{V}_{DR}(f)$ is guaranteed to be unbiased for $V_{DR}$ *even if the outcome model is incorrect*. In other words, DR-CPO is robust to misspecification of $\widehat{g}$, as the IPW term in its value function corrects for any bias from the predicted outcomes. This means that rather than requiring a randomized dataset for training, $\widehat{g}$ can reasonably be learned over *any* direct outcome dataset, including those where the causal relationship between the text and the outcome may be confounded.

Therefore, in addition to learning from the experimental data $X \sim P^R, Y \sim P_y^R$, a model optimized with DR-CPO will be able to further leverage the generative language model $f^0$ to learn from unlimited unlabeled text $\widetilde{X} \sim P^{f^0}$ with predicted outcomes $\widehat{g}(\widetilde{X})$. This can reduce the variance of the value function estimator.

**Proposition 4.5.** *If $\widehat{g}$ is fit on a separate sample, then conditional on $\widehat{g}$, $n\left(Var\left(\widehat{V}_{IPW}(f)\right) - Var\left(\widehat{V}_{DR}(f)\right)\right)$ is equal to*

$$Var\left(\frac{P^f(X)}{P^R(X)}g(X)\right) - Var\left(\frac{P^f(X)}{P^R(X)}\left(g(X) - \widehat{g}(X)\right)\right)$$
$$- \frac{n}{m}Var\left(\frac{P^f(\widetilde{X})}{P^{f^0}(\widetilde{X})}\widehat{g}(\widetilde{X})\right)$$

Proposition 4.5 shows the difference in the variances of the IPW and DR value function estimators, scaled by the sample size $n$ of the crowdsourced data to highlight asymptotic differences. This difference indicates that $Var(\widehat{V}_{DR}(f)) < Var(\widehat{V}_{IPW}(f))$ subject to two conditions: (i) the number of Monte Carlo samples drawn from $f^0$ is much larger than the sample size of the crowdsourced data, i.e., $m \gg n$; and (ii) $\widehat{g}(x)$ has *some* additional predictive power compared to a constant model.

Condition (i) limits the component of the variance difference that arises due to Monte Carlo error from taking $m$ samples from the reference language model $f^0$. Then the main comparison is the difference between the variance of the expected outcome $g(x)$, and the variance of the *prediction* error for the model. Consequently, under condition (ii), we can expect the variance of $\widehat{V}_{DR}$ to be lower than the variance of $\widehat{V}_{IPW}$ (e.g., if $(\widehat{g}(x) - g(x))^2 < g(x)^2$).

We note that in a different data setting where the true $P^R$ is unknown (e.g., no crowdsourced or randomized data is available), DR-CPO can still solve the true causal problem if all confounders are known in the dataset used to train $\widehat{g}$. Controlling for all confounders allows the outcome model to learn the correct causal relationship between the text and the outcome. Since such an outcome model is correct, it fulfills the condition $\widehat{g}(X) = g(X)$, and it follows from Theorem 4.4 that the DR-CPO objective $\widehat{V}_{DR}$ is an unbiased estimator for the true value function $V(f)$ even if $P^R$ is not randomized.

In practice, the full set of confounders is rarely known (particularly for text data). However, a good $\widehat{g}$ can still likely be learned if a large amount of clean data exists to train the outcome model. In this case, a well-estimated outcome model $\widehat{g}$ can still help bias-correct mis-estimation of $\widehat{P}^R(X)$.

## 4.3 RELATIONSHIP TO EXISTING APPROACHES

Elements of (DR-)CPO are reflected in two of the most prominent existing language model optimization approaches: RLHF and DPO.

First, notice that the outcome modeling term $V_{out}$ is itself a way of identifying $V(f)$ from the observed data. Outcome modeling relies entirely on the predictive model $\widehat{g}$ and unlabeled texts $\widetilde{X} \sim P^{f^0}$ and therefore does not require an experimental dataset $\mathcal{D}_R$. However, if $\widehat{g}(X)$ is not a good outcome model, then $\widehat{V}_{out}(f)$ will be biased with respect to $V(f)$. Reward models trained on confounded data may be misspecified, since they can only capture the relationship between the text and the response—and not the confounders that have additionally influenced the response. While these issues are remedied if the confounding is also fully modeled, confounders are extremely difficult to measure fully in text data.

Optimization of $V_{out}(f)$ is closely related to RLHF in the direct outcome setting via proximal policy optimization (PPO) and REINFORCE [Stiennon et al., 2020, Williams, 1992, Ahmadian et al., 2024]. In particular, the PPO loss term can be seen as a version of outcome modeling in which the reward model is trained on paired completion data. Under these conditions, the RLHF reward model is analogous to the outcome model $g$, and the PPO loss is mathematically equivalent to $V_{out}(f)$ (details in Appendix B). Because in practice PPO often optimizes both the policy network and the value function network, $V_{out}(f)$—which optimizes only the policy network—can also be seen as similar to a REINFORCE objective in which learning occurs over Monte Carlo samples drawn from a fixed language model $f^0$.

Likewise, DPO shares similarities with CPO. Rather than relying on reward or outcome modeling, both DPO and CPO fine-tune a language model for human feedback by directly using a preference dataset, which greatly reduces their computational overhead. The DPO objective is similar to $V_{IPW}$ in that it directly increases the likelihood of texts corresponding to desired outcomes through importance weighting. However, because it uses paired data, DPO increases the density ratio between preferred and non-preferred examples, while CPO directly increases the probability of texts with desired outcomes and decreases the probability of texts with non-desired outcomes. Given a paired data setting, the DPO objective could possibly be recovered from $V_{IPW}$; we leave this derivation for future work.

# 5 EXPERIMENTS

We conduct evaluations to empirically assess the effectiveness of CPO and DR-CPO in optimizing language models for human preferences on direct outcome data, and we examine the doubly robust properties of DR-CPO under confounding.

## 5.1 DATASETS

To evaluate optimization on direct outcome data, we consider three crowdsourced or randomized experimental datasets in which human annotators provided numerical responses to texts.

**Hate Speech** *(binary outcome).* The Hate Speech dataset [Qian et al., 2019] consists of comments from the social media sites Reddit and Gab. Comments span a wide variety of subjects, including daily life, personal relationships, politics, and finance. Outcomes are collected via crowdsourcing and indicate whether the annotator perceives the comment to be hate speech. The Reddit comments are chosen from subreddits where hate speech is more common, and Gab is a platform where users sometimes migrate after being blocked from other social media sites. The optimization goal for this dataset is to generate texts that are *less* hateful on average.

We include this dataset to evaluate optimization under conditions where (i) texts are topically diverse and (ii) outcomes are relatively simple to judge.

**Hong Kong** *(scalar outcome).* The Hong Kong dataset [Fong and Grimmer, 2021] consists of texts concerning the Hong Kong democracy protests of 2019-2020. These texts are loosely based on speeches made about Hong Kong during U.S. Congressional sessions at the time of the protests. Outcomes are collected via a randomized experiment and indicate on a scale of 0-100 to what extent the respondent thinks that the U.S. should support Hong Kong during this time, after reading the text. The texts are programmatically constructed: for each text, 2 or 3 text attributes are randomly chosen out of 7 (e.g., *commitment*, *bravery*, *mistreatment*). Short passages corresponding to each attribute are then randomly chosen from a pool of about 20 to construct the text. The optimization goal for this dataset is to generate texts with *high* outcomes on average.

We include this dataset to evaluate optimization under more challenging conditions where (i) texts all concern a single topic and (ii) outcomes are complex, noisy, and difficult for a model to predict.

**Confounded** *(scalar outcome).* The Confounded dataset is a version of the Hong Kong dataset where we have induced confounding. Concretely, it is identical to the Hong Kong dataset with the exception of its outcome, which is the negation of the original outcome (e.g., an outcome of 60 in the Hong Kong dataset becomes an outcome of -60 in the Confounded dataset). This inverts the relationship between the text and the outcome: texts originally associated with high outcomes in the Hong Kong dataset are now associated with low outcomes, and vice versa. Intuitively, negating the outcome is akin to introducing a strong confounding or corrupting factor where readers who tend to read texts that *should* have high outcomes also tend to have very *negative* responses to those texts (e.g., "haters" brigading a popular celebrity's posts).

We include this dataset with the realistic expectation that text data is often confounded. While CPO by construction uses only randomized data, confounded data can pose a threat to approaches that use non-randomized data to train their outcome models. Therefore, it is necessary to evaluate how approaches like DR-CPO and OO-RLHF fare when their outcome models are trained on confounded data.

## 5.2 IMPLEMENTATION

**Evaluation.** To evaluate how well optimization for human preferences has occurred, we consider two separate metrics. As our primary evaluation, we use a text preference framework in which a reader is asked to choose the better

(with respect to the outcome) of a pair of texts generated by two different methods. Using GPT-4 as a proxy for human annotators, we compare pairs of *(method, baseline)* completions for the same prompt; across all pairs, we compute method *win rates* and compute 95% confidence intervals. Since the datasets used for these experiments contain one text per sample rather than a prompt and a completion, we create prompts on the evaluation set by truncating each text to a random length.

The full input provided to GPT-4 for each dataset can be found in Appendix C.2. We validate the use of GPT-4 as an annotator with a human study, which we describe in further detail in Section 6.1.

To supplement our text preference evaluation, we additionally assess the effectiveness of each method in maximizing expected reward (i.e., the value function). In particular, we focus on the reward under the DR-CPO objective, which is the most efficient estimate of the reward.

**Methods.** We evaluate language models optimized using **CPO** and **DR-CPO**. As our baselines, we consider language models that have been fine-tuned on texts from each of the task datasets (**FT**), as well as models optimized using the outcome modeling value function $V_{out}(f)$. Since—as we discuss in Section 4.3—the $V_{out}(f)$ objective is mathematically equivalent to the RLHF objective, we refer to this baseline as **OO-RLHF** (offline outcome RLHF).

We use Llama 2 7B [Touvron et al., 2023] as our base language model and fine-tune with low-rank adaptation (LoRA) [Hu et al., 2022]. All optimizations are applied after fine-tuning on text from the task dataset. Because only a tiny fraction of parameters are updated under LoRA, we note that we expect relatively small changes in performance under optimization.

Our outcome models for the Hate Speech dataset are also based on Llama 2 7B. For the Hong Kong outcome model, LLMs perform poorly on the dataset's complex outcomes, particularly given the dataset's smaller size. Instead, we use a standard linear regression model over features extracted using the Empath lexicon [Fast et al., 2016].

Implementation details can be found in Appendix C.1.

**Choice of $f^0$.** When optimizing with DR-CPO or OO-RLHF, any generative language model may be used as $f^0$, the fixed language model from which texts are drawn as input to the outcome model. One key consideration is whether $f^0$ should be a pre-trained model or whether it should be a model that has been fine-tuned on text relevant to the task—for instance, the randomized experiment dataset $\mathcal{D}_R$.

In practice, we choose a pre-trained model as $f^0$ to leverage the diversity of texts such models tend to generate. If $\widehat{g}(X)$ is a good outcome model, then predicted outcomes on these texts will still be close to the true outcomes, and DR-CPO

and OO-RLHF will benefit from outcome modeling.

**Choice of $\widehat{P}^R$.** In Section 4.1, we mention that although the distribution of texts $P^R(X)$ under the randomized experiment is known, it can be helpful empirically to instead compute an estimated $\widehat{P}^R(X)$. This is generally due to the fact that the *sample* probability of each text $X$ may not actually be equal to its theoretical probability merely by chance [Hájek, 1971, Särndal et al., 2003].

We find this to be the case in our experiments, and so we use $\widehat{P}^R(X)$ estimated from a Llama 2 7B model fine-tuned on $\mathcal{D}_R$ in our CPO and DR-CPO implementations.

# 6 RESULTS AND DISCUSSION

## 6.1 GPT-4 ANNOTATION VALIDITY

| Annotator 1 | Annotator 2 | Fleiss' $\kappa$ |
|---|---|---|
| Human | Human | 0.170 |
| Human | GPT-4 | 0.219 |
| Human majority | GPT-4 | 0.192 |

Table 1: Agreement rates of human annotators and GPT-4 when asked to choose preferred texts with respect to target outcomes. We examine inter-human agreement, human-GPT-4 agreement, and agreement between a majority vote of human annotators and GPT-4.

Following the precedent set by Rafailov et al. [2023], we conduct a human study to assess the validity of GPT-4 as an annotator when choosing between pairs of texts for a preferred outcome. Across 200 randomly sampled examples from the Hong Kong dataset, we show human annotators *(method, baseline)* completion pairs and ask them to choose the better of the two with respect to the outcome. We compute agreement between each human annotator, as well as agreement between each human annotator and GPT-4. Agreement is measured through Fleiss' $\kappa$ [Fleiss, 1971], a common metric for agreement among multiple raters.

We use the online research platform Prolific[7] to conduct our human study. To prevent annotator fatigue, examples are annotated in batches of 20. We recruit a total of 30 annotators for an average of 3 annotators per example and a total of 600 annotations.

Across three comparisons—human-human, majority vote-human, and human-GPT-4—we find that GPT-4 exhibits a similar or better level of agreement with human annotators as human annotators do with each other (Table 1).

We conclude that GPT-4 is a reasonable surrogate for human annotators.

---
[7]https://www.prolific.co/

| Dataset | Method | DR-CPO reward | CPO reward | OO-RLHF reward |
| --- | --- | --- | --- | --- |
| Hate Speech | FT | 0.219 | -0.024 | 0.242 |
| Hate Speech | OO-RLHF | 0.244 | -0.006 | 0.249 |
| Hate Speech | CPO | 0.230 | -0.012 | 0.242 |
| Hate Speech | DR-CPO | **0.245** | **-0.005** | **0.250** |
| Hong Kong | FT | 24.505 | -0.256 | 24.761 |
| Hong Kong | OO-RLHF | 22.256 | -2.753 | 25.009 |
| Hong Kong | CPO | **25.969** | **1.254** | 24.715 |
| Hong Kong | DR-CPO | 25.162 | 0.269 | **24.893** |

Table 2: (Stabilized) expected reward under the DR-CPO, CPO, and OO-RLHF objectives. Expected reward under the DR-CPO objective is the most efficient estimate of the reward and therefore the most reliable metric. On the Hate Speech dataset, DR-CPO achieves the highest expected reward, while on the Hong Kong dataset, CPO achieves the highest reward.

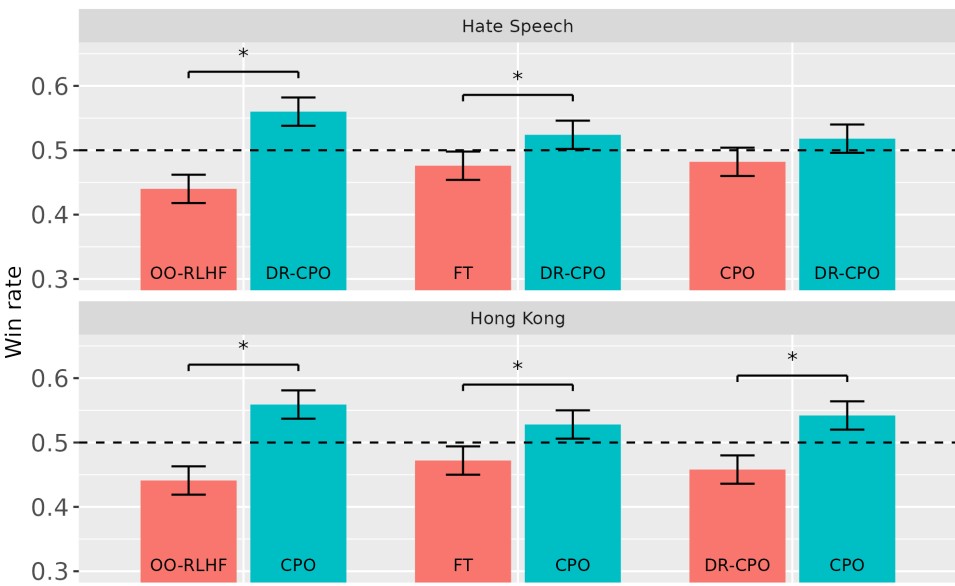

Figure 1: CPO and DR-CPO win rates against OO-RLHF, FT, and one another other. A win rate exceeding 0.5 indicates that the named method outperforms the competing method with respect to the target outcome. The error bars correspond to 95% confidence intervals, and asterisks (*) mean that the win rate difference between the two methods is statistically significant at the 95% confidence level.

## 6.2 OUTCOME OPTIMIZATION

Over 2000 test prompts for each dataset, we evaluate CPO and DR-CPO win rates against baselines OO-RLHF and FT (and against one another). Under this evaluation framework, a win rate greater than 0.5 indicates that the method outperforms its competitor. We report our results in Figure 1. Additional results are found in Appendix C.3.

With the same 2000 prompts, we further evaluate expected reward[8] under each optimization method. We reiterate that our optimization goal is to maximize the expected outcome

or reward of texts generated by the language model. We report our results in Table 2.

On the Hate Speech dataset, we observe that DR-CPO outperforms both the OO-RLHF and FT baselines with respect to win rate. Against both baselines, DR-CPO's win rate is statistically significant at the 95% confidence level, with the lower bound of its 95% confidence intervals falling above 0.5. DR-CPO also achieves the highest expected reward across all objectives. These results indicate that using DR-CPO, language models successfully learn human preferences for less hateful text from direct outcomes—and do so better than competing baselines.

DR-CPO additionally appears to outperform CPO on the

---

[8]Rewards are stabilized to prevent numerical underflow and overflow.

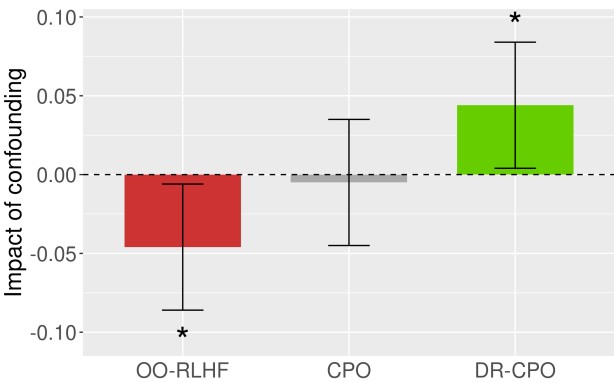

Figure 2: Impact of outcome model confounding (measured in win rate difference divided by 2) on OO-RLHF, CPO, and DR-CPO. A negative impact indicates that confounding hurts the performance of the method. The error bars correspond to 95% confidence intervals, and asterisks (*) mean that the win rate difference is statistically significant at the 95% confidence level.

Hate Speech dataset, though its win rate is just shy of statistical significance. This result supports our theoretical result (Proposition 4.5) that the addition of an outcome model to the CPO objective can improve optimization by reducing variance. Moreover, the success of DR-CPO over OO-RLHF provides further evidence for the benefits of the doubly robust formulation, wherein a good $P^R$ can help to correct bias from an outcome modeling approach.

On the Hong Kong dataset, we likewise observe that CPO outperforms both the OO-RLHF and FT baselines with respect to both win rate and expected reward under the DR-CPO objective. In this setting, CPO achieves statistically significant win rates against both baselines, with the lower bound of its 95% confidence intervals falling above 0.5.

Here, in contrast to the Hate Speech dataset, CPO also outperforms DR-CPO, which suggests that CPO can be very strong under conditions where $P^R$ is well controlled or can be estimated very well—as is the case for the Hong Kong dataset, where texts are not only randomly assigned to annotators but programmatically generated from random attributes. An additional factor may be the difficulty of modeling the complex continuous outcome of the Hong Kong dataset from its artificially constructed, homogeneous texts. Under such conditions, an outcome model may introduce noise rather than help with bias correction (see Proposition 4.5 for a discussion of the objectives' precision), and so CPO may enjoy empirical advantages over DR-CPO. The challenge of outcome modeling is additionally reflected in the underperformance of OO-RLHF relative to even FT, providing evidence for how outcome modeling alone can be insufficient for language model optimization.

## 6.3 DOUBLE ROBUSTNESS UNDER CONFOUNDING

In Figure 2, we examine the impact of outcome model confounding on OO-RLHF, CPO, and DR-CPO. For our confounded condition, we train both the DR-CPO and OO-RLHF outcome models on the Confounded dataset. As DR-CPO also uses randomized data in its IPW term, we optimize this portion of the objective on the (unconfounded) Hong Kong dataset. For each method, we then compute win rate between a model that has been optimized under these confounding conditions and a model that has been optimized on unconfounded data using the same method.

CPO is constructed to use only randomized data and does not have an outcome model, so it is exclusively optimized on the (unconfounded) Hong Kong dataset. However, we compare two separately optimized models to account for randomness in the optimization process.

Win rates are computed across 600 pairs for each method. Additional results are found in Appendix C.3.

In these experiments, we find that DR-CPO remains robust under confounding in the outcome model, while OO-RLHF degrades significantly. OO-RLHF experiences an impact to win rate that is significantly negative, while CPO experiences no impact (as expected, because it does not use an outcome model) and DR-CPO experiences a positive impact. Exploring this last result is an avenue for future work.

These results highlight one of the core strengths of DR-CPO and one of the core shortcomings of outcome modeling approaches. As long as a small crowdsourced dataset is available to act as $P^R$, DR-CPO can use large amounts of confounded data to learn its outcome model and still retain its unbiasedness guarantees—all while reducing variance relative to CPO. An exclusively outcome model-based approach like OO-RLHF, on the other hand, becomes biased under these conditions. Even under aggressive confounding, with a worst-case outcome model that has been trained on completely negated data, DR-CPO is not compromised, while OO-RLHF is.

We reiterate that because they are optimized (at least partially) on randomized experimental data, CPO and DR-CPO are *causal* approaches. Taken together, our results emphasize the importance of an optimization framework that maintains the causal relationship between text and outcome.

## 7 CONCLUSION

In this paper, we explore language model optimization for human preferences from direct outcome datasets, in which each sample consists of a text and the reader's numerical response. We first posit that language model optimization should be viewed as a causal problem to ensure that the

model correctly learns the relationship between the text and the outcome, and we define conditions under which this causal relationship can be guaranteed. Following this, we introduce CPO, a method that solves an unbiased surrogate objective for the causal language optimization problem—and improve upon it with the doubly robust DR-CPO, which reduces the variance of the CPO objective while retaining provably strong guarantees on bias. Finally, we empirically demonstrate the effectiveness of (DR-)CPO in optimizing state-of-the-art LLMs for human preferences on direct outcome data, and we validate the robustness of DR-CPO under difficult confounding conditions.

To our knowledge, our work is the first to approach LLM optimization as a causal inference problem—from which an importance-weighted solution naturally follows—as well as the first to propose a doubly robust methodology for LLM optimization. These theoretical contributions and results open the door to a wide range of data, human preferences, and optimization goals that language models can learn using (DR-)CPO.

Several natural lines of future research follow from this work. For instance, (DR-)CPO may benefit empirically from entropy regularization techniques that are common in policy optimization. Additionally, the performance gains in our evaluations may have been limited by the small proportion of parameters updated under LoRA, and so it would be useful to explore whether experiments with commercial-grade computational resources have the potential to yield much larger improvements. Finally, future work may wish to extend DR-CPO to the paired completion data setting, as the bias guarantees and variance reduction of a doubly robust approach can also be useful for paired data.

## Acknowledgements

This material is based upon work partially supported by the National Institutes of Health (awards R01MH125740, R01MH132225, and R21MH130767). Victoria Lin is supported by a Meta Research PhD Fellowship. Any opinions, findings, conclusions, or recommendations expressed in this material are those of the author(s) and do not necessarily reflect the views of the sponsors, and no official endorsement should be inferred.

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

# Optimizing Language Models for Human Preferences is a Causal Inference Problem (Supplementary Material)

Victoria Lin[1]  Eli Ben-Michael[1]  Louis-Philippe Morency[1]

[1]Carnegie Mellon University

# A  TECHNICAL RESULTS

## A.1  IDENTIFYING $V(f)$

*Proof.* Proof of Proposition 4.1

We can show that $V(f) = \mathbb{E}_{Y(\cdot)\sim\mathcal{G}}[\mathbb{E}_{X\sim P^f}[Y(X)]] = \mathbb{E}_{Y\sim P_y^R}[\mathbb{E}_{X\sim P^R}[\frac{P^f(X)}{P^R(X)}Y]]$.

$$V(f) = \mathbb{E}_{Y(\cdot)\sim\mathcal{G}}[\mathbb{E}_{X\sim P^f}[Y(X)]]$$

$$= \mathbb{E}_{Y(\cdot)\sim\mathcal{G}}\left[\sum_{x\in\mathcal{X}} P^f(x)Y(x)\right]$$

$$= \mathbb{E}_{Y(\cdot)\sim\mathcal{G}}\left[\sum_{x\in\mathcal{X}} P^R(x)\frac{P^f(x)}{P^R(x)}Y(x)\right]$$

$$= \mathbb{E}_{Y(\cdot)\sim\mathcal{G}}\left[\mathbb{E}_{X\sim P^R}\left[\frac{P^f(X)}{P^R(X)}Y(X)\right]\right]$$

$$= \mathbb{E}_{Y\sim P_y^R}\left[\mathbb{E}_{X\sim P^R}\left[\frac{P^f(X)}{P^R(X)}Y\right]\right]$$

$\square$

*Proof.* Proof of Proposition 4.3

We can show that $V(f) = \mathbb{E}_{Y(\cdot)\sim\mathcal{G}}[\mathbb{E}_{X\sim P^f}[Y(X)]] = \mathbb{E}_{Y\sim P_y^R}[\mathbb{E}_{X\sim P^R}[\frac{P^f(X)}{P^R(X)}(Y-g(X))]] + \mathbb{E}_{X\sim P^f}[g(X)]$.

$$V(f) = \mathbb{E}_{Y\sim P_y^R}\left[\mathbb{E}_{X\sim P^R}\left[\frac{P^f(X)}{P^R(X)}Y\right]\right] \quad\text{(Proposition 4.1)}$$

$$= \mathbb{E}_{Y\sim P_y^R}\left[\mathbb{E}_{X\sim P^R}\left[\frac{P^f(X)}{P^R(X)}(Y-g(X)+g(X))\right]\right]$$

$$= \mathbb{E}_{Y\sim P_y^R}\left[\mathbb{E}_{X\sim P^R}\left[\frac{P^f(X)}{P^R(X)}(Y-g(X))\right]\right] + \mathbb{E}_{Y\sim P_y^R}\left[\mathbb{E}_{X\sim P^R}\left[\frac{P^f(X)}{P^R(X)}g(X)\right]\right]$$

$$= \mathbb{E}_{Y\sim P_y^R}\left[\mathbb{E}_{X\sim P^R}\left[\frac{P^f(X)}{P^R(X)}(Y-g(X))\right]\right] + \mathbb{E}_{X\sim P^R}\left[\frac{P^f(X)}{P^R(X)}g(X)\right]$$

$$= \mathbb{E}_{Y \sim P_y^R} \left[ \mathbb{E}_{X \sim P^R} \left[ \frac{P^f(X)}{P^R(X)} (Y - g(X)) \right] \right] + \sum_{x \in \mathcal{X}} P^R(x) \frac{P^f(x)}{P^R(x)} g(x)$$

$$= \mathbb{E}_{Y \sim P_y^R} \left[ \mathbb{E}_{X \sim P^R} \left[ \frac{P^f(X)}{P^R(X)} (Y - g(X)) \right] \right] + \sum_{x \in \mathcal{X}} P^f(x) g(x)$$

$$= \mathbb{E}_{Y \sim P_y^R} \left[ \mathbb{E}_{X \sim P^R} \left[ \frac{P^f(X)}{P^R(X)} (Y - g(X)) \right] \right] + \mathbb{E}_{X \sim P^f} [g(X)]$$

$\square$

## A.2 UNBIASEDNESS PROOFS AND VARIANCE DIFFERENCE

*Proof.* Proof of Theorem 4.2

We can show that $\mathbb{E}_{Y(\cdot) \sim \mathcal{G}}[\mathbb{E}_X[\widehat{V}_{IPW}(f)]] = \mathbb{E}_{Y(\cdot) \sim \mathcal{G}}[\mathbb{E}_{X \sim P^f}[Y(X)]] = V(f)$ when $P^R$ is known.

$$\mathbb{E}_{Y(\cdot) \sim \mathcal{G}}[\mathbb{E}_{X \sim P^R}[\widehat{V}_{IPW}(f)]] = \mathbb{E}_{Y(\cdot) \sim \mathcal{G}} \left[ \mathbb{E}_{X \sim P^R} \left[ \frac{1}{n} \sum_{i=1}^{n} \frac{P^f(X_i)}{P^R(X_i)} Y_i \right] \right]$$

$$= \frac{1}{n} \sum_{i=1}^{n} \mathbb{E}_{Y(\cdot) \sim \mathcal{G}} \left[ \mathbb{E}_{X \sim P^R} \left[ \frac{P^f(X_i)}{P^R(X_i)} Y_i \right] \right]$$

$$= \mathbb{E}_{Y(\cdot) \sim \mathcal{G}} \left[ \mathbb{E}_{X \sim P^R} \left[ \sum_{x \in \mathcal{X}} \frac{P^f(x)}{P^R(x)} Y(x) \mathbb{1}\{X = x\} \right] \right]$$

$$= \mathbb{E}_{Y(\cdot) \sim \mathcal{G}} \left[ \sum_{x \in \mathcal{X}} \frac{P^f(x)}{P^R(x)} Y(x) \underbrace{\mathbb{E}_{X \sim P^R}[\mathbb{1}\{X = x\}]}_{P^R(x)} \right]$$

$$= \mathbb{E}_{Y(\cdot) \sim \mathcal{G}} \left[ \sum_{x \in \mathcal{X}} P^f(x) Y(x) \right]$$

$$= \mathbb{E}_{Y(\cdot) \sim \mathcal{G}}[\mathbb{E}_{X \sim P^f}[Y(X)]]$$

$$= V(f)$$

$\square$

*Proof.* Proof of Theorem 4.4

We can show that $\mathbb{E}_{Y(\cdot) \sim \mathcal{G}}[\mathbb{E}_X[\widehat{V}_{DR}(f)]] = \mathbb{E}_{Y(\cdot) \sim \mathcal{G}}[\mathbb{E}_{X \sim P^f}[Y(X)]] = V(f)$ under one of two conditions: $\widehat{P}^R(X) = P^R(X)$ (i.e., $P^R$ is known) or (2) $\widehat{g}(X) = g(X) = \mathbb{E}_{Y(\cdot) \sim \mathcal{G}}[Y(X)]$ (i.e., $g(X)$ is known).

First, we rewrite $\mathbb{E}_{Y(\cdot) \sim \mathcal{G}}[\mathbb{E}_X[\widehat{V}_{DR}(f)]]$:

$$\mathbb{E}_{Y(\cdot) \sim \mathcal{G}}[\mathbb{E}_{X, \widetilde{X}}[\widehat{V}_{DR}(f)]] = \mathbb{E}_{Y(\cdot) \sim \mathcal{G}} \left[ \mathbb{E}_{X, \widetilde{X}} \left[ \frac{1}{n} \sum_{i=1}^{n} \frac{P^f(X_i)}{\widehat{P}^R(X_i)} (Y_i - \widehat{g}(X_i)) + \frac{1}{m} \sum_{j=1}^{m} \frac{P^f(\widetilde{X}_j)}{P^{f^0}(\widetilde{X}_j)} \widehat{g}(\widetilde{X}_j) \right] \right]$$

$$= \mathbb{E}_{Y(\cdot) \sim \mathcal{G}} \left[ \mathbb{E}_{X \sim P^R} \left[ \frac{1}{n} \sum_{i=1}^{n} \frac{P^f(X_i)}{\widehat{P}^R(X_i)} (Y_i - \widehat{g}(X_i)) \right] \right]$$

$$+ \mathbb{E}_{Y(\cdot) \sim \mathcal{G}} \left[ \mathbb{E}_{\widetilde{X} \sim P^{f^0}} \left[ \frac{1}{m} \sum_{j=1}^{m} \frac{P^f(\widetilde{X}_j)}{P^{f^0}(\widetilde{X}_j)} \widehat{g}(\widetilde{X}_j) \right] \right]$$

$$= \frac{1}{n} \sum_{i=1}^{n} \mathbb{E}_{Y(\cdot) \sim \mathcal{G}} \left[ \mathbb{E}_{X \sim P^R} \left[ \frac{P^f(X_i)}{\widehat{P}^R(X_i)} (Y_i - \widehat{g}(X_i)) \right] \right]$$

$$+ \frac{1}{m} \sum_{j=1}^{m} \mathbb{E}_{Y(\cdot)\sim\mathcal{G}} \left[ \mathbb{E}_{\widetilde{X}\sim P^{f^0}} \left[ \frac{P^f(\widetilde{X}_j)}{P^{f^0}(\widetilde{X}_j)} \widehat{g}(\widetilde{X}_j) \right] \right]$$

Case (1) $\widehat{P}^R(X) = P^R(X)$:

Rewriting the first term,

$$\frac{1}{n} \sum_{i=1}^{n} \mathbb{E}_{Y(\cdot)\sim\mathcal{G}} \left[ \mathbb{E}_{X\sim P^R} \left[ \frac{P^f(X_i)}{\widehat{P}^R(X_i)} (Y_i - \widehat{g}(X_i)) \right] \right] = \frac{1}{n} \sum_{i=1}^{n} \mathbb{E}_{Y(\cdot)\sim\mathcal{G}} \left[ \mathbb{E}_{X\sim P^R} \left[ \frac{P^f(X_i)}{P^R(X_i)} (Y_i - \widehat{g}(X_i)) \right] \right]$$

$$= \mathbb{E}_{Y(\cdot)\sim\mathcal{G}} \left[ \mathbb{E}_{X\sim P^R} \left[ \sum_{x\in\mathcal{X}} \frac{P^f(x)}{P^R(x)} (Y(x) - \widehat{g}(x)) \mathbb{1}\{X = x\} \right] \right]$$

$$= \mathbb{E}_{Y(\cdot)\sim\mathcal{G}} \left[ \sum_{x\in\mathcal{X}} \frac{P^f(x)}{P^R(x)} (Y(x) - \widehat{g}(x)) \underbrace{\mathbb{E}_{X\sim P^R}[\mathbb{1}\{X = x\}]}_{P^R(x)} \right]$$

$$= \mathbb{E}_{Y(\cdot)\sim\mathcal{G}} \left[ \sum_{x\in\mathcal{X}} P^f(x) (Y(x) - \widehat{g}(x)) \right]$$

$$= \mathbb{E}_{Y(\cdot)\sim\mathcal{G}}[\mathbb{E}_{X\sim P^f}[Y(X) - \widehat{g}(X)]]$$

$$= \mathbb{E}_{Y(\cdot)\sim\mathcal{G}}[\mathbb{E}_{X\sim P^f}[Y(X)]] - \mathbb{E}_{Y(\cdot)\sim\mathcal{G}}[\mathbb{E}_{X\sim P^f}[\widehat{g}(X)]]$$

Rewriting the second term,

$$\frac{1}{m} \sum_{j=1}^{m} \mathbb{E}_{Y(\cdot)\sim\mathcal{G}} \left[ \mathbb{E}_{\widetilde{X}\sim P^{f^0}} \left[ \frac{P^f(\widetilde{X}_j)}{P^{f^0}(\widetilde{X}_j)} \widehat{g}(\widetilde{X}_j) \right] \right] = \mathbb{E}_{Y(\cdot)\sim\mathcal{G}} \left[ \mathbb{E}_{\widetilde{X}\sim P^{f^0}} \left[ \sum_{x\in\mathcal{X}} \frac{P^f(x)}{P^{f^0}(x)} \widehat{g}(x) \mathbb{1}\{\widetilde{X} = x\} \right] \right]$$

$$= \mathbb{E}_{Y(\cdot)\sim\mathcal{G}} \left[ \sum_{x\in\mathcal{X}} \frac{P^f(x)}{P^{f^0}(x)} \widehat{g}(x) \underbrace{\mathbb{E}_{\widetilde{X}\sim P^{f^0}}[\mathbb{1}\{\widetilde{X} = x\}]}_{P^{f^0}(x)} \right]$$

$$= \mathbb{E}_{Y(\cdot)\sim\mathcal{G}} \left[ \sum_{x\in\mathcal{X}} P^f(x) \widehat{g}(x) \right]$$

$$= \mathbb{E}_{Y(\cdot)\sim\mathcal{G}}[\mathbb{E}_{X\sim P^f}[\widehat{g}(X)]]$$

Then we have

$$\mathbb{E}_{Y(\cdot)\sim\mathcal{G}}[\mathbb{E}_X[\widehat{V}_{DR}(f)]] = \mathbb{E}_{Y(\cdot)\sim\mathcal{G}}[\mathbb{E}_{X\sim P^f}[Y(X)]] - \mathbb{E}_{Y(\cdot)\sim\mathcal{G}}[\mathbb{E}_{X\sim P^f}[\widehat{g}(X)]] + \mathbb{E}_{Y(\cdot)\sim\mathcal{G}}[\mathbb{E}_{X\sim P^f}[\widehat{g}(X)]]$$

$$= \mathbb{E}_{Y(\cdot)\sim\mathcal{G}}[\mathbb{E}_{X\sim P^f}[Y(X)]]$$

$$= V(f)$$

Case (2) $\widehat{g}(X) = g(X) = E_{Y(\cdot)\sim\mathcal{G}}[Y(X)]$:

Rewriting the first term,

$$\frac{1}{n} \sum_{i=1}^{n} \mathbb{E}_{Y(\cdot)\sim\mathcal{G}} \left[ \mathbb{E}_{X\sim P^R} \left[ \frac{P^f(X_i)}{\widehat{P}^R(X_i)} (Y_i - \widehat{g}(X_i)) \right] \right] = \frac{1}{n} \sum_{i=1}^{n} \mathbb{E}_{Y(\cdot)\sim\mathcal{G}} \left[ \mathbb{E}_{X\sim P^R} \left[ \frac{P^f(X_i)}{\widehat{P}^R(X_i)} (Y_i - g(X_i)) \right] \right]$$

$$= \mathbb{E}_{Y(\cdot)\sim\mathcal{G}} \left[ \mathbb{E}_{X\sim P^R} \left[ \sum_{x\in\mathcal{X}} \frac{P^f(x)}{\widehat{P}^R(x)} (Y(x) - g(x)) \mathbb{1}\{X = x\} \right] \right]$$

$$= \mathbb{E}_{Y(\cdot)\sim\mathcal{G}} \left[ \sum_{x\in\mathcal{X}} \frac{P^f(x)}{\widehat{P}^R(x)} (Y(x) - g(x)) \underbrace{\mathbb{E}_{X\sim P^R}[\mathbb{1}\{X = x\}]}_{P^R(x)} \right]$$

$$= \mathbb{E}_{Y(\cdot) \sim \mathcal{G}} \left[ \sum_{x \in \mathcal{X}} \frac{P^f(x) P^R(x)}{\widehat{P}^R(x)} (Y(x) - g(x)) \right]$$

$$= \sum_{x \in \mathcal{X}} \frac{P^f(x) P^R(x)}{\widehat{P}^R(x)} \mathbb{E}_{Y(\cdot) \sim \mathcal{G}}[Y(x) - g(x)]$$

$$= \sum_{x \in \mathcal{X}} \frac{P^f(x) P^R(x)}{\widehat{P}^R(x)} (\mathbb{E}_{Y(\cdot) \sim \mathcal{G}}[Y(x)] - g(x))$$

$$= \sum_{x \in \mathcal{X}} \frac{P^f(x) P^R(x)}{\widehat{P}^R(x)} \cdot 0$$

$$= 0$$

Rewriting the second term,

$$\frac{1}{m} \sum_{j=1}^{m} \mathbb{E}_{Y(\cdot) \sim \mathcal{G}} \left[ \mathbb{E}_{\widetilde{X} \sim P^{f^0}} \left[ \frac{P^f(\widetilde{X}_j)}{P^{f^0}(\widetilde{X}_j)} \widehat{g}(\widetilde{X}_j) \right] \right] = \frac{1}{m} \sum_{j=1}^{m} \mathbb{E}_{Y(\cdot) \sim \mathcal{G}} \left[ \mathbb{E}_{\widetilde{X} \sim P^{f^0}} \left[ \frac{P^f(\widetilde{X}_j)}{P^{f^0}(\widetilde{X}_j)} g(\widetilde{X}_j) \right] \right]$$

$$= \mathbb{E}_{Y(\cdot) \sim \mathcal{G}} \left[ \mathbb{E}_{\widetilde{X} \sim P^{f^0}} \left[ \sum_{x \in \mathcal{X}} \frac{P^f(x)}{P^{f^0}(x)} g(x) \mathbb{1}\{\widetilde{X} = x\} \right] \right]$$

$$= \mathbb{E}_{Y(\cdot) \sim \mathcal{G}} \left[ \sum_{x \in \mathcal{X}} \frac{P^f(x)}{P^{f^0}(x)} g(x) \underbrace{\mathbb{E}_{\widetilde{X} \sim P^{f^0}}[\mathbb{1}\{\widetilde{X} = x\}]}_{P^{f^0}(x)} \right]$$

$$= \mathbb{E}_{Y(\cdot) \sim \mathcal{G}} \left[ \sum_{x \in \mathcal{X}} P^f(x) g(x) \right]$$

$$= \sum_{x \in \mathcal{X}} P^f(x) g(x)$$

$$= \mathbb{E}_{X \sim P^f}[g(X)]$$

$$= \mathbb{E}_{X \sim P^f}[E_{Y(\cdot) \sim \mathcal{G}}[Y(X)]]$$

$$= \mathbb{E}_{Y(\cdot) \sim \mathcal{G}}[\mathbb{E}_{X \sim P^f}[Y(X)]]$$

Then we have

$$\mathbb{E}_{Y(\cdot) \sim \mathcal{G}}[\mathbb{E}_X[\widehat{V}_{DR}(f)]] = 0 + \mathbb{E}_{Y(\cdot) \sim \mathcal{G}}[\mathbb{E}_{X \sim P^f}[Y(X)]]$$

$$= \mathbb{E}_{Y(\cdot) \sim \mathcal{G}}[\mathbb{E}_{X \sim P^f}[Y(X)]]$$

$$= V(f)$$

$$\square$$

*Proof.* Proof of Proposition 4.5

First, we compute the variance of $\widehat{V}_{IPW}(f)$, where $X_i \sim P^R$ and $Y_i \sim P_y^R$ i.i.d.

$$\text{Var}\left(\widehat{V}_{IPW}(f)\right) = \text{Var}\left(\frac{1}{n} \sum_{i=1}^{n} \frac{P^f(X_i)}{P^R(X_i)} Y_i\right)$$

$$= \frac{1}{n^2} \sum_{i=1}^{n} \text{Var}\left(\frac{P^f(X_i)}{P^R(X_i)} Y_i\right)$$

$$= \frac{1}{n} \text{Var}\left(\frac{P^f(X)}{P^R(X)} Y\right)$$

$$= \frac{1}{n} \mathbb{E}\left[\text{Var}\left(\frac{P^f(X)}{P^R(X)} Y \mid X\right)\right] + \frac{1}{n} \text{Var}\left(\mathbb{E}\left[\frac{P^f(X)}{P^R(X)} Y \mid X\right]\right)$$

$$= \frac{1}{n}\mathbb{E}\left[\frac{P^f(X)^2}{P^R(X)^2}\text{Var}\left(Y \mid X\right)\right] + \frac{1}{n}\text{Var}\left(\frac{P^f(X)}{P^R(X)}\mathbb{E}\left[Y \mid X\right]\right)$$

$$= \frac{1}{n}\mathbb{E}\left[\frac{P^f(X)^2}{P^R(X)^2}\text{Var}\left(Y \mid X\right)\right] + \frac{1}{n}\text{Var}\left(\frac{P^f(X)}{P^R(X)}g(X)\right)$$

where we have used that under randomization $\mathbb{E}[Y \mid X = x] = \mathbb{E}_{Y(\cdot)\sim\mathcal{G}}[Y(x)] = g(x)$.

For the variance of $\widehat{V}_{DR}(f)$, we first note that if $\widehat{g}$ is fit on a separate, independent sample, we have that

$$\text{Var}\left(\widehat{V}_{DR}(f)\right) = \text{Var}\left(\frac{1}{n}\sum_{i=1}^{n}\frac{P^f(X_i)}{P^R(X_i)}(Y_i - \widehat{g}(X_i)) + \frac{1}{m}\sum_{j=1}^{m}\frac{P^f(\widetilde{X}_j)}{P^{f^0}(\widetilde{X}_j)}\widehat{g}(\widetilde{X}_j)\right)$$

$$= \frac{1}{n^2}\sum_{i=1}^{n}\text{Var}\left(\frac{P^f(X_i)}{P^R(X_i)}(Y_i - \widehat{g}(X_i))\right) + \frac{1}{m^2}\sum_{j=1}^{m}\text{Var}\left(\frac{P^f(\widetilde{X}_j)}{P^{f^0}(\widetilde{X}_j)}\widehat{g}(\widetilde{X}_j)\right)$$

$$= \frac{1}{n}\underbrace{\text{Var}\left(\frac{P^f(X)}{P^R(X)}(Y - \widehat{g}(X))\right)}_{(*)} + \frac{1}{m}\text{Var}\left(\frac{P^f(\widetilde{X})}{P^{f^0}(\widetilde{X})}\widehat{g}(\widetilde{X})\right)$$

where $X \sim P^R$, $Y \sim P_y^R$, and $\widetilde{X} \sim P^{f^0}$. Now notice that

$$(*) = \mathbb{E}\left[\text{Var}\left(\frac{P^f(X)}{P^R(X)}(Y - \widehat{g}(X)) \mid X\right)\right] + \text{Var}\left(\mathbb{E}\left[\frac{P^f(X)}{P^R(X)}(Y - \widehat{g}(X)) \mid X\right]\right)$$

$$= \mathbb{E}\left[\frac{P^f(X)^2}{P^R(X)^2}\text{Var}\left(Y \mid X\right)\right] + \text{Var}\left(\frac{P^f(X)}{P^R(X)}(g(X) - \widehat{g}(X))\right)$$

$\square$

## A.3  EQUIVALENCE OF $V_{out}(f)$

We can show that our rewriting of $V_{out}(f)$ is equivalent to our original definition:

$$V_{out}(f) = \mathbb{E}_{\widetilde{X}\sim P^{f^0}}\left[\frac{P^f(X)}{P^{f^0}(X)}g(X)\right]$$

$$= \sum_{x\in\mathcal{X}} P^{f^0}(x)\frac{P^f(x)}{P^{f^0}(x)}g(x)$$

$$= \sum_{x\in\mathcal{X}} P^f(x)g(x)$$

$$= E_{X\sim P^f}[g(X)]$$

## B  PARALLELS BETWEEN RLHF AND OPTIMIZATION OF $V_{out}(f)$

The loss function under RLHF is typically computed through proximal policy optimization (PPO):

$$\mathcal{L}(\theta,\phi) = \mathcal{L}_{\text{policy}}^{PPO}(\theta) + c_1\mathcal{L}_{\text{value}}^{PPO}(\phi) - c_2\mathcal{L}_{\text{entropy}}^{PPO}(\theta)$$

where $\mathcal{L}_{\text{value}}^{PPO}(\phi)$ and $\mathcal{L}_{\text{entropy}}^{PPO}(\theta)$ are regularization terms and $\mathcal{L}_{\text{policy}}^{PPO}(\theta)$ is the *policy loss*. Letting $p$ denote the prompt, $c$ denote the completion, and $r$ denote the reward model, we consider only the policy loss without any stability tricks like clipping. $\pi_\theta$ is the probability under the policy being optimized, while $\pi_{\theta_0}$ is the probability under a reference policy (often the starting policy or the policy at the previous step).

$$\mathcal{L}_{PPO}(\theta) = \mathbb{E}\left[\frac{\pi_\theta(c|p)}{\pi_{\theta_0}(c|p)} \cdot r(p,c)\right]$$

We can see the equivalence between $\pi_\theta(c|p)$ and $P^f(X)$, $\pi_{\theta_0}(c|p)$ and $P^{f^0}(X)$, and $r(p, c)$ and $g(X)$; substituting these terms renders $\mathcal{L}_{PPO}$ equal to $V_{out}(f)$.

# C  EXPERIMENTS

## C.1  MODEL DETAILS AND HYPERPARAMETERS

Our language models were implemented using the HuggingFace `transformers` library (version 4.32.1), with Llama 2 7B weights stored locally. For each language model optimization method, we trained for 3 epochs with a batch size of 4, maximum gradient norm of 1.0, and learning rate of $5 \times 10^{-5}$, with all other hyperparameters set to Trainer class defaults. We used the HuggingFace `peft` library for LoRA and set our LoraConfig parameters as follows: rank 8, alpha 8, dropout probability 0.1, with all other parameters remaining as default. Custom loss functions for CPO, DR-CPO, and OO-RLHF were implemented using PyTorch (`torch` version 2.0.1).

For the Hong Kong dataset outcome model, we used `scikit-learn` (version 1.3.0) to implement our linear regression. All hyperparameters were set to their default values. Empath features were extracted using the `empath` library.[1]

## C.2  GPT-4 WIN RATE PROMPTS

Here, we include the inputs provided to GPT-4 for each dataset to obtain its preferences between texts generated by different methods. We accessed GPT-4 through the OpenAI API (`gpt-4-1106-preview`). Prompt formats are adapted from those described in Rafailov et al. [2023]. The order of texts is random for each evaluation. For a *(method, baseline)* comparison, *method* corresponds to text A 50% of the time and text B 50% of the time.

### C.2.1  Hate Speech prompt

```
Hate speech is a direct attack on an individual based on protected characteristics: race,
    ethnicity, national origin, religious affiliation, sexual orientation, caste, sex,
    gender, gender identity, and serious disease or disability.

Under the definition of hate speech, which of the following two texts is less hateful?

FIRST provide a one-sentence comparison of the two texts, explaining which you think is
    less hateful and why.

SECOND, on a new line, state only "A" or "B" to indicate your choice. Your response should
     use the format:
Comparison: <one-sentence comparison and explanation>
Preferred: <"A" or "B">

A: <text A>

B: <text B>
```

### C.2.2  Hong Kong and Confounded prompt

```
In March 2019, the Hong Kong government proposed a bill that would have allowed certain
    criminal suspects to be extradited to mainland China for trial. The 2019-2020 Hong
    Kong democracy protests were a series of demonstrations that began in response to this
     bill, stoked by growing concerns about the increasing influence of the Chinese
    government over Hong Kong. Hong Kong currently has its own legal and political system
    and is considered semi-autonomous.
```

---

[1] https://github.com/Ejhfast/empath-client

```
Which of the following two texts better convinces you that the U.S. government should
    support the Hong Kong democracy protesters?

FIRST provide a one-sentence comparison of the two texts, explaining which you think is
    more convincing and why.

SECOND, on a new line, state only "A" or "B" to indicate your choice. Your response should
    use the format:
Comparison: <one-sentence comparison and explanation>
Preferred: <"A" or "B">

A: <text A>

B: <text B>
```

## C.3 ADDITIONAL RESULTS

| | Hong Kong | | Hate Speech | |
| --- | --- | --- | --- | --- |
| | CPO win rate | DR-CPO win rate | CPO win rate | DR-CPO win rate |
| FT | **0.528\* [0.506, 0.550]** | 0.477 [0.455, 0.499] | 0.517 [0.495, 0.539] | **0.524\* [0.502, 0.546]** |
| CPO | - | 0.441 [0.419, 0.463] | - | 0.518 [0.496, 0.540] |
| OO-RLHF | **0.542\* [0.520, 0.564]** | 0.482 [0.460, 0.504] | **0.538\* [0.516, 0.560]** | **0.560\* [0.538, 0.582]** |
| DR-CPO | 0.559\* [0.537, 0.581] | - | 0.482 [0.460, 0.504] | - |

Table 3: CPO and DR-CPO win rates against OO-RLHF, FT, and each other. A win rate exceeding 0.5 indicates that the named method outperforms the competing method with respect to the target outcome. Win rates are computed across 2000 pairs for each method combination.

| | Unconfounded win rate over confounded |
| --- | --- |
| OO-RLHF | **0.546\* [0.506, 0.586]** |
| DR-CPO | **0.456\* [0.416, 0.496]** |
| CPO | 0.505 [0.465, 0.545] |

Table 4: Win rates with outcome models trained on unconfounded data (Hong Kong) vs. confounded data (Confounded). A win rate exceeding 0.5 indicates that the method+unconfounded outcome model outperforms the method+confounded outcome model with respect to the target outcome—in other words, that confounding hurts the method. Win rates are computed across 600 pairs for each method combination.

We include the full set of CPO and DR-CPO win rates against OO-RLHF and FT and against each other (Table 3). We also include raw win rates from the confounding experiments, specifically win rates of unconfounded methods over confounded methods (Table 4). We briefly discuss comparisons that did not appear in the main body of the paper.

On the Hate Speech dataset, we observe that CPO—like DR-CPO—also outperforms both the OO-RLHF and FT baselines. Against OO-RLHF, CPO's win rate is statistically significant at the 95% confidence level, while its win rate against FT falls slightly short of statistical significance.

On the Hong Kong dataset, we find that DR-CPO performs comparably to OO-RLHF but falls short against the other methods. We attribute this to possible difficulty in learning the outcome model itself; this is further evidenced by the strong performance of CPO, which does not use an outcome model. As we mention in the main results, learning a strong outcome model on the Hong Kong dataset may be challenging, as its texts read somewhat artificially due to their programmatic construction.