# OpenReview forum: "Optimizing Language Models for Human Preferences is a Causal Inference Problem"
_auai.org/UAI/2024/Conference — UAI 2024 poster_

### Official Review · Reviewer_ZKZY · 2024-03-06

**Q2-1 Originality-Novelty:** 3
**Q2-2 Correctness-Technical Quality:** 3
**Q2-5 Clarity Of Writing:** 3

**Q1 Summary And Contributions:**

This paper explores language model optimization for human preferences using direct outcome datasets, where each sample consists of a text and a numerical outcome reflecting the reader's response. The paper proposes viewing language model optimization as a causal problem to ensure that the model accurately learns the relationship between the text and the outcome. The authors formalize this causal language optimization problem and introduce a method called CPO that solves an unbiased surrogate objective for the problem. They further enhance CPO with doubly robust CPO, which reduces the variance of the surrogate objective while maintaining strong guarantees on bias.

**Q2-3 Extent To Which Claims Are Supported By Evidence:**

3: Good: the main claims are supported by convincing evidence (in the form of adequate experimental evaluation, proofs, (pseudo-)code, references, assumptions).

**Q2-4 Reproducibility:**

3: Good: key resources (e.g. proofs, code, data) are available and key details (e.g. proofs, experimental setup) are sufficiently well-described for competent researchers to confidently reproduce the main results.

**Q3 Main Strengths:**

The paper's approach of formulating LLM optimization as a causal inference problem and introducing the DR-CPO method is intriguing. It bridges the gap between language model training and statistical inference, offering a fresh perspective on LLM optimization.

The paper is well-organized, and its theoretical foundation is robust. However, I find the approach slightly impractical.

**Q4 Main Weakness:**

This paper presents simple experiments to validate its theoretical approach, acknowledging the challenge of evaluating a theory-driven method in extensive experiments.

The paper offers a fresh perspective by framing the optimization problem from a causal standpoint. Despite the novelty of this approach, the contribution of the technique, particularly in double machine learning, is well-developed.

**Q5 Detailed Comments To The Authors:**

I appreciate this paper and believe it would greatly benefit from acceptance, as its intriguing ideas deserve to be shared with a wider audience.

**Q9 Complying With Reviewing Instructions:**

Yes

---

> ### Author Rebuttal · Authors · 2024-04-09
>
> Thank you for your feedback and kind remarks!
>
> **[“Simple experiments”]** We agree that theory-driven methods can be difficult to evaluate. Therefore, we looked to other papers for reference when choosing our evaluations, and our tasks are comparable to other theory-driven papers on LLM optimization (e.g., Stiennon et al., 2020; Rafailov et al, 2024). Given the nascence of the field, it remains an open question how best to do experimental evaluations when optimizing LLMs for human preferences, and we think it is a good line of future research.
>
> **[“[T]he contribution of the technique, particularly in double machine learning, is well-developed”]** As the reviewer points out—and as we mention in our related works— doubly robust objectives are used in a variety of machine learning tasks. Adapting the idea of doubly robust estimation to solve new problems has been an important line of machine learning research.
>
> In our case—again as the reviewer mentions—the novelty of our work comes from adapting the principles of importance weighting and doubly robust estimation to better optimize LLMs. Moreover, we believe that drawing the connection between LLM optimization and causal inference is itself an important contribution, and we provide extensive mathematical derivations that support the theoretical validity of our methods. To our knowledge, our work is the first to examine the parallels between LLM optimization and causal inference, as well as the first to propose a doubly robust methodology for LLM optimization.
>
> Stiennon, N., Ouyang, L., Wu, J., Ziegler, D., Lowe, R., Voss, C., ... & Christiano, P. F. (2020). Learning to summarize with human feedback. Advances in Neural Information Processing Systems, 33, 3008-3021.
>
> Rafailov, R., Sharma, A., Mitchell, E., Manning, C. D., Ermon, S., & Finn, C. (2024). Direct preference optimization: Your language model is secretly a reward model. Advances in Neural Information Processing Systems, 36.

---

### Official Review · Reviewer_rRjr · 2024-03-08

**Q2-1 Originality-Novelty:** 3
**Q2-2 Correctness-Technical Quality:** 3
**Q2-5 Clarity Of Writing:** 2

**Q1 Summary And Contributions:**

The motivation, as stated in the paper, is that in text-outcome datasets, there is typically confounding present: E.g., "hateful" readers are more likely to engage with hateful content in the first place, which means that such text may be evaluated more positively than a random reader would react. The goal is then to optimize LMs for human preferences while considering this confounding problem.

The paper does not seem to solve the confounding problem with a technical contribution directly. Instead, it focuses on using crowdsourced data in which the human evaluators are randomized -- thus mitigating the confounding aspect. Prior work also usually uses crowdsourced data, so this is not a differentiating aspect.

The actual contribution, in my view, is thus not of a causal nature: it is to optimize LMs on outcome data in the first place, which has some slight differences from optimizing LMs on *preference* data. Thus, humans making choices between two samples are in this work replaced by humans giving a numerical rating.

**Q2-3 Extent To Which Claims Are Supported By Evidence:**

3: Good: the main claims are supported by convincing evidence (in the form of adequate experimental evaluation, proofs, (pseudo-)code, references, assumptions).

**Q2-4 Reproducibility:**

3: Good: key resources (e.g. proofs, code, data) are available and key details (e.g. proofs, experimental setup) are sufficiently well-described for competent researchers to confidently reproduce the main results.

**Q3 Main Strengths:**

The mathematical results seem correct to me. I also think (as far as I am aware) that all three methods that they create (CPO, (DR-)CPO, and OO-RLHF) have not been studied before in the LM context, i.e., I believe it was not attempted before to train LMs to maximize human evaluation scores, instead of respecting preference comparisons. (I may be wrong about this, but trust that the authors have faithfully described prior work)

**Q4 Main Weakness:**

1. The main weakness, in my view, is that this paper does not make a technical contribution to the causal problem that is posed. I would be very happy for the authors to clarify if I misunderstood anything, and will change my score if they can convince me that I'm wrong. Otherwise, it seems to me that the motivation and technical contribution are essentially orthogonal to each other.

To explain: It is true that "in the wild", there is confounding between texts and who reads a text. But it is already common practice in RLHF to use crowdsourced datasets where human evaluators are randomized, so that this problem does not occur. In this paper, the authors do not differ from this practice and assume crowdsourced datasets. There is no technical contribution that explains how to solve the causal problem when crowdsourcing is NOT possible.

To be clear, the authors *do* claim that one of their methods solves the causal problem even under actual confounding -- namely, DR-CPO. However, in the mathematical derivations of this method, I find no indication of why it should not also be susceptible to confounding problems, and thus, it is hard for me to believe the empirical results in Figure 2.


2. The second main weakness concerns their evaluation baseline. The authors choose OO-RLHF as the baseline upon which they intend to improve. However, as far as I can tell, this method is also created by the authors themselves, and it is unclear to me why it should serve as a baseline, instead of another method to compare e.g. against supervised fine-tuning. If this was done, then it would result that supervised fine-tuning actually works better than OO-RLHF according to their own evaluation.

**Q5 Detailed Comments To The Authors:**

I did at first not understand where the distribution P^R is coming from, and especially why you are able to *evaluate* it (if it indeed comes from human-written text). Later, you clarify that you evaluate this distribution approximately by fine-tuning an LM. Maybe this could be clarified earlier.

For the outcome model g^, I would like to see more details on how it is trained. Is it trained before, or alongside f? Is it's correctness validated in any way?

The authors write "Vanilla CPO is not affected by confounding because it does not use an outcome model". I think this is wrong. It seems to me that the optimization objective of CPO is in the middle of the left column in p. 4, and that when there is confounding in the data, Y_i is not a causal outcome of X_i anymore. The authors should either clarify or remove the claim.

**Q9 Complying With Reviewing Instructions:**

Yes

---

> ### Author Rebuttal · Authors · 2024-04-09
>
> Thank you for your detailed, thoughtful feedback!
>
> **[Technical contribution]** Our intent is not to frame the use of crowdsourced datasets as a novel contribution. Rather:
> 1. Our *theoretical* contribution is to formulate LLM optimization as a causal inference problem. To our knowledge, we are the first to do this. While crowdsourced preference datasets are sometimes used in RLHF, they are not chosen with the explicit intent of establishing a causal relationship between text and outcome. Moreover, preference datasets used to train LLMs often are *not* randomized (e.g., user feedback to ChatGPT outputs) and therefore may be confounded.
> 2. **Our *methodological* contributions are two methods—CPO and DR-CPO—that solve unbiased surrogates for the true causal problem *using crowdsourced data.*** Both methods require at least some unconfounded data to guarantee unbiasedness. CPO uses *only* crowdsourced data, but DR-CPO can *additionally* leverage confounded data in its outcome modeling term.
>
>  	Re: DR-CPO and confounding: DR-CPO is robust to confounding *in the data used to train the outcome model*. As stated in Theorem 4.4, as long as either $P^R$ or $g$ is known, the DR-CPO objective is unbiased for the true causal problem (proven in Appendix A). Since $P^R$ is known when the data is crowdsourced (see “$P^R$ details” below), then even if $g$ is learned on confounded data and therefore biased, the overall objective is still unbiased. Thus, DR-CPO can be optimized on large amounts of potentially confounded data as long as a small crowdsourced dataset is available to act as $P^R$. We show this empirically in Figure 2.
>
>  	We will update the paper to emphasize clearly that DR-CPO requires crowdsourced data.
>
> Finally, if crowdsourcing is not possible, a causal relationship between text and outcome cannot be established unless all confounders are known. This is very rarely the case for text data. Since crowdsourced direct outcome data is abundant in NLP, relying on a small amount of randomized data is quite practical compared to having to identify all confounders.
>
> If we *do* know all confounders, then it is possible to use DR-CPO to solve the true causal problem *even without crowdsourced/randomized data.* This was not discussed in the paper but came up afterward due to the reviewer’s comment. Consider the setting where $P^R$ is *not* randomized, but we know all confounders. If the outcome model controls for all confounders in addition to the text, it should be correct (i.e., $\hat{g}(X)=g(X)$), and it follows from Theorem 4.4 that the the DR-CPO objective $\hat{V}_{DR}(f)$ is an unbiased estimator for the true value function $V(f)$ even if $P^R$ is not randomized. In the next version of the paper, we will add a discussion of this scenario.
>
> **[Evaluation baseline]** We considered both OO-RLHF and the supervised fine-tuned model (FT) to be baselines but may not have stated this clearly enough in the paper. We compared our methods against both OO-RLHF and FT in Fig 1, where we observed that (DR-)CPO significantly outperforms both baselines (\* denotes statistical significance at 95% confidence). We consider OO-RLHF a baseline because it is not inherently causal (and is also an ablation of DR-CPO). OO-RLHF can indeed be worse than FT, as seen in Fig 1; this reflects how outcome modeling alone can be insufficient for LLM optimization.
>
> **[$P^R$ details]** If $P^R$ is a randomized experiment, then the probability $P^R(X)$ of any text $X \sim P^R$ is known according to the randomization probability. For instance, if texts are uniformly assigned to $n$ readers at random, where one reader reads one text, then theoretically $P^R(X)=\frac{1}{n}$ for all $X \sim P^R$. As we note in the paper, sometimes it is more statistically efficient to estimate $\hat{P}^R$ than using the theoretical distribution, which is why we estimate $\hat{P}^R$ from a fine-tuned LM.
>
> We will include this earlier in the paper when $P^R$ is first introduced.
>
> **[Training outcome model]** The outcome model $\hat{g}$ is trained on a held-out set of $P^R$ before $f$ is trained. On a validation split of the Hong Kong dataset, the outcome model had MSE=658 (compared to mean-prediction MSE=750), while on the Hate Speech dataset, the outcome model had accuracy=94% and F1=0.75. We will publicly release our code on GitHub after the review process is completed, so these details and many more (e.g., hyperparameters) will be available.
>
> **[Vanilla CPO and confounding]** We refer specifically to confounding *in the data used to train the outcome model*; in our paper, we define $P^R$ as a crowdsourced dataset and therefore never confounded. CPO is not affected by confounding in the data used to train the outcome model because it does not use an outcome model. Fig 2 is meant to show that DR-CPO remains performant even when the outcome model is confounded, while OO-RLHF does not. We will update the paper and figure captions to clarify the nature of the confounding.

---

### Official Review · Reviewer_XMhC · 2024-03-13

**Q2-1 Originality-Novelty:** 3
**Q2-2 Correctness-Technical Quality:** 3
**Q2-5 Clarity Of Writing:** 4

**Q10 Ethical Concerns:**

No ethical concerns were raised unknown to the RLHF domain.

**Q1 Summary And Contributions:**

The paper introduces a novel method that allows using human-labeled data (e.g., scored answers to a question) in RLHF without the need to have paired completions using well-known counterfactual methods in offline learning, such as IPW and DR.

**Q2-3 Extent To Which Claims Are Supported By Evidence:**

3: Good: the main claims are supported by convincing evidence (in the form of adequate experimental evaluation, proofs, (pseudo-)code, references, assumptions).

**Q2-4 Reproducibility:**

1: Poor: key details (e.g. proof sketches, experimental setup) are incomplete/unclear, or key resources (e.g. proofs, code, data) are unavailable.

**Q3 Main Strengths:**

The paper is well-written, the problem is clearly motivated, and the method assumptions are well-discussed.
The analogies of the method to RLHF and DPO techniques help to understand key differences.

**Q4 Main Weakness:**

Although I commend the authors for being thorough in the experiment methodology, I miss the description of the hyperparameters used for training, which are crucial for any future reproducibility.

**Q5 Detailed Comments To The Authors:**

If I understand correctly, according to Appendix B, optimization of V_{out}(f) is done only on the policy network, hence being more similar to vanilla policy gradient methods, such as REINFORCE. I understand that PPO is more popular in practice, and there are undeniable similarities, but I think it is misleading to label it this way, especially as REINFORCE has been getting some attention recently [Ahmadian, 2024].

Ahmadian, A., Cremer, C., Gallé, M., Fadaee, M., Kreutzer, J., Pietquin, O., Üstün, A., & Hooker, S. (2024). Back to Basics: Revisiting REINFORCE Style Optimization for Learning from Human Feedback in LLMs (arXiv:2402.14740). arXiv. https://doi.org/10.48550/arXiv.2402.14740

**Q9 Complying With Reviewing Instructions:**

Yes

---

> ### Author Rebuttal · Authors · 2024-04-09
>
> Thank you for your feedback and for pointing out many of the strengths of our work!
>
> **[Hyperparameters and code]** We agree that reproducibility and transparency of experiments, including source code and hyperparameters, is important. We will include these details in the next version of the paper.
>
> As optimization was done using low-rank adaptation (LoRA), we trained all our models—including outcome models—with the HuggingFace PEFT library. We set our LoRA parameters as follows: rank=8, alpha=8, dropout probability=0.1, with all other parameters remaining as default. Our HuggingFace Trainer arguments were as follows: epochs=3, batch size=4, maximum gradient norm=1.0, learning rate=5e-5, with all other parameters again remaining as default.
>
> For the Hong Kong outcome model only, we found LLMs to perform poorly on the dataset’s complex outcomes. We hypothesized that this could be due to the smaller size of the Hong Kong dataset, so for this model only, we instead used a standard linear regression model over lexicon features extracted using [Empath](https://github.com/Ejhfast/empath-client). The linear regression model was fit using the sklearn library and used all default parameters.
>
> All code will be uploaded publicly to GitHub once the review process is completed, and a link will be included in the paper. If the reviewer wishes to see the source code before then, it is attached to our submission as supplementary material.
>
> **[REINFORCE vs. PPO]** Thank you for raising this distinction—this is a great point. We will update the paper to indicate that our outcome modeling objective more closely aligns with methods like REINFORCE.

---

### Official Review · Reviewer_QNXu · 2024-03-13

**Q2-1 Originality-Novelty:** 2
**Q2-2 Correctness-Technical Quality:** 2
**Q2-5 Clarity Of Writing:** 3

**Q1 Summary And Contributions:**

This paper investigates the alignment of language models with human preferences using direct outcome datasets, consisting of texts and their numerical outcomes (reader responses). The authors argue for treating language model optimization as a causal problem, modelling relationships between texts and outcomes. They introduce Causal Preference Optimization (CPO) and its extension, Doubly Robust CPO (DR-CPO), which aims to optimize language models by solving an unbiased surrogate objective for this problem, with DR-CPO additionally reducing variance while maintaining bias guarantees. Their empirical evaluations demonstrate the effectiveness of these methods in optimizing language models for human preferences under both standard and confounding conditions.

**Q2-3 Extent To Which Claims Are Supported By Evidence:**

2: Fair: the main claims are somewhat supported by evidence (but the experimental evaluation may be weak, or does not match entirely with the claims, important baselines may be missing, proofs contain important ideas but lack rigor, algorithmic details are only discussed superficially, references are imprecise, assumptions are not sufficiently motivated or explicated, etc.).

**Q2-4 Reproducibility:**

3: Good: key resources (e.g. proofs, code, data) are available and key details (e.g. proofs, experimental setup) are sufficiently well-described for competent researchers to confidently reproduce the main results.

**Q3 Main Strengths:**

- It is interesting to frame the language model optimization as a causal inference problem, demonstrating the causal relationships between texts and outcomes.
- The paper is well-written and easy to follow.

**Q4 Main Weakness:**

- While the identification and estimation of value function V(f) through importance weighting principles is a good choice, the novelty of this method is limited.

- The approach's reliance on high-quality direct outcome datasets might limit its applicability in scenarios where such datasets are scarce, biased, or difficult to obtain.

**Q5 Detailed Comments To The Authors:**

Please refer to the weaknesses mentioned above.

**Q9 Complying With Reviewing Instructions:**

Yes

---

> ### Author Rebuttal · Authors · 2024-04-09
>
> Thank you for your review! We address your comments below.
>
> **[Novelty of importance weighting]** We believe the methodological novelty of our work comes from adapting existing principles from causal inference to *better optimize LLMs*. We think that drawing the connection between LLM optimization and causal inference is itself an important contribution, and we provide extensive mathematical derivations that support the theoretical validity of our methods. Moreover, we develop solutions based on not only importance weighting but also *doubly robust* estimation. To our knowledge, our work is the first to examine the parallels between LLM optimization and causal inference—from which an importance-weighted solution naturally follows—as well as the first to propose a doubly robust methodology for LLM optimization.
>
> **[Reliance on direct outcome datasets]** We agree that high-quality direct outcome datasets are not available for every setting. However, datasets consisting of texts associated with a crowdsourced label are much more common in NLP than the paired preference datasets that are required for current LLM optimization methods like RLHF and DPO (for instance, thousands of direct outcome datasets are hosted on HuggingFace’s Datasets hub, compared to fewer than 50 paired preference datasets). Direct outcome datasets are also much more common than having the resources to collect one’s own data in an online reinforcement learning setting, which would be the other alternative. Therefore, we view the use of direct outcome datasets as an expansion, rather than limitation, on applicability.

---

### Official Review · Reviewer_Da3i · 2024-03-23

**Q2-1 Originality-Novelty:** 3
**Q2-2 Correctness-Technical Quality:** 3
**Q2-5 Clarity Of Writing:** 2

**Q10 Ethical Concerns:**

None.

**Q1 Summary And Contributions:**

The paper presents a solution for fine tuning based on a causal method, namely double robust causal preference optimization. The authors present a theoretical analysis of the characteristics of the proposed method and an experimental evaluation using annotated datasets.

**Q2-3 Extent To Which Claims Are Supported By Evidence:**

2: Fair: the main claims are somewhat supported by evidence (but the experimental evaluation may be weak, or does not match entirely with the claims, important baselines may be missing, proofs contain important ideas but lack rigor, algorithmic details are only discussed superficially, references are imprecise, assumptions are not sufficiently motivated or explicated, etc.).

**Q2-4 Reproducibility:**

2: Fair: key resources (e.g. proofs, code, data) are unavailable but key details (e.g. proof sketches, experimental setup) are sufficiently well-described for an expert to confidently reproduce the main results.

**Q3 Main Strengths:**

- The idea of introducing causality to the problem of language fine-tuning is interesting and promising.

- The proposed approach appears to be based on a strong mathematical foundation. The authors report exhaustive results concerning the characteristics of the proposed methods.

**Q4 Main Weakness:**

- The reviewer has concerns about the evaluation of the method. The reviewer understands that it is very hard to evaluate these types of approaches. However, the authors only report data about win results and not absolute performance values. It is very difficult to determine how good the proposed approaches are for example.

-The authors should report some commentary about the different behavior that is observed for the different datasets using the proposed technique. It is very difficult to understand why certain methods appear to perform better than others for certain datasets for example. This appears to be essential, given the fact it might be possible to argue that the evaluation is somehow limited to two quite specific datasets. It is difficult to say if the proposed method will generalize to other datasets or not.

- The reviewer would like to point out that the difference between the proposed methods in Figure 2 and 3 is quite close at the end. Someone might argue that the relatively small difference in performance does not justify the deployment of the proposed technique. The reviewer was not able to understand the trade-off in terms of accuracy/complexity offered by the proposed method. This is an example about which the reviewer might want to comment about, since it appears to be rather central in my opinion.

- The idea of studying the impact of confounding is very interesting. However, the reviewer was not able to clearly understand the characteristics of the dataset used for this experiment. For this reason, the reviewer found the assessment of the results presented in Figure 2 not really easy to interpret.

- The authors used essentially datasets about hate speech. The reviewer wonders how the proposed approach might be extended to other types of datasets and if the performances that are presented in this study are somehow a function of the specific situation of hate speech, where, in general, the judgement of a certain situation, the extraction of potential counterfactuals, etc. might be relatively "simple" and not noisy.

**Q5 Detailed Comments To The Authors:**

Additional comments/questions

1. It would be good if the authors could comment on the type of information that is present in the Hong Kong dataset and how/why finetuning is necessary for it.

2. Could you please comment on the different results for different methods/datasets? How should we interpret the results shown in Figure 1 and 2?

3. Could you please comment on the possible extension of the proposed methods to different types of text (i.e., not involving hate speech?). Do you have any additional results, etc for other datasets?

**Q9 Complying With Reviewing Instructions:**

Yes

---

> ### Author Rebuttal · Authors · 2024-04-09
>
> Thank you for your thorough feedback and for highlighting the mathematical strengths of our work! We will update the paper with the following discussions and results.
>
> **[Evaluation - absolute performance]** Following Rafailov et al. (2023), we report (stabilized) expected reward under the DR-CPO, CPO, and OO-RLHF objectives. Expected reward under the DR-CPO objective is the most efficient estimate of the reward and therefore the key metric.
> | Dataset 	|  Method | DR-CPO reward | CPO reward | OO-RLHF reward |
> |-------------|:-------:|:-------------:|:----------:|:--------------:|
> | Hong Kong (HK)  | FT  	|    	24.505 | 	-0.256 |     	24.761 |
> | HK  | CPO 	|    	**25.969** |  	**1.254** |     	24.715 |
> | HK  | OO-RLHF |    	22.256 | 	-2.753 |     	**25.009** |
> | HK   | DR-CPO  |    	25.162 |  	0.269 |     	24.893 |
> | Hate Speech (HS) | FT  	|     	0.219 | 	-0.024 |      	0.242 |
> | HS | CPO 	|     	0.230 | 	-0.012 |      	0.242 |
> | HS | OO-RLHF |     	0.244 | 	-0.006 |      	0.249 |
> | HS | DR-CPO  |     	**0.245** | 	**-0.005** |      	**0.250** |
>
> Consistent with the win rates, DR-CPO has the highest expected reward on the HS dataset, while CPO has the highest expected reward on HK.
>
> Rafailov et al. Direct preference optimization: Your language model is secretly a reward model. NeurIPS 2023.
>
> **[Generality of datasets: ”[A]uthors used essentially datasets about hate speech”; Q3: “[D]ifferent types of text”]** To clarify, while one dataset concerns hate speech (HS), we also evaluated extensively on the very different Hong Kong (HK) dataset:
> - Texts: HK texts are based on US Congressional speeches, contain themes like patriotism and duty, and are programmatically constructed (Sec 5.1). HS contains natural texts from social media spaces with toxic speech and covers many topics.
> - Outcomes: HK outcomes are a 0-100 rating of support for the 2019-2020 Hong Kong protests after reading the text. These are complex, noisy, and much harder for a model to predict than the binary HS outcome.
>
> **[Behaviors for different datasets: ”[W]hy certain methods appear to perform better than others for certain datasets”; Q2: “[I]nterpret the results shown in [Figs] 1 and 2”]** Parts of Figs 1 and 2 could be more clearly explained. The error bars are 95% confidence intervals, and in Fig 1, \* means win rate difference is statistically significant at 95% confidence. We will update our captions accordingly.
>
> In Fig 1, one key difference is that DR-CPO performs best on the HS dataset (significantly better than OO-RLHF/FT), while CPO performs best on the HK dataset (significantly better than all other methods). This may be because the binary HS outcome is easier to predict than the continuous HK outcome, so the HS outcome model is better-calibrated and including it in DR-CPO improves performance. For HK the outcome model may instead introduce noise (see Prop 4.5 for a theoretical comparison of the objectives’ precision).
>
> For Fig 2, see paragraph 2 of “Characteristics of confounding dataset” below.
>
> **[Small differences in performance”]** Due to limited computational resources, we optimized our models using low-rank adaptation in which only 0.06% of weights are updated. As changes to the model are small, small performance differences are also expected. We consider our empirical results an important proof-of-concept, and we believe performance gains may be much larger with commercial-grade computational resources.
>
> **[Accuracy/complexity tradeoffs]**
> - CPO improves performance over fine-tuning (FT) at little computational cost. Each optimization step requires only a single pass over the data, with no generation.
> - DR-CPO improves performance over FT at similar computational cost to paired reward modeling approaches like RLHF (because they require generating texts and outcomes) *while conferring additional robustness guarantees.*
>
> **[Confounding dataset characteristics]** The Confounded dataset is the HK dataset with the outcome negated: texts originally associated with high outcomes in HK are now associated with low outcomes, and vice versa. This is akin to a confounding or corrupting factor where readers who tend to read texts that *should* have high outcomes also tend to have very *negative* responses to those texts (e.g. “haters” brigading a popular celebrity’s posts).
>
> Fig 2 compares the win rate when the outcome model is trained on the Confounded dataset vs. the (unconfounded) HK dataset, while $P^R$ is fixed as the HK dataset. OO-RLHF (which only uses the confounded outcome model) is negatively impacted by confounding, but CPO (which does not use the confounded data) and DR-CPO (which uses the unconfounded $P^R$ to bias-correct the confounded outcome model) are robust to confounding.
>
> **[Q1: “[C]omment on… the [HK] dataset and how/why finetuning is necessary”]** Please see “Generality of datasets.” Fine-tuning is necessary for both datasets because their texts are very different from the pre-training data of our base LLM, Llama2.

---

### Meta-Review · Area_Chair_N2tk · 2024-04-16

On the positive side, I'd cut and paste what a reviewer mentioned in the discussion: Considering the popularity of RLHF with LLMs these days, there are lots of new papers in the field. This is one of the better papers that I've seen. It systematically introduces new ideas, bridges a few separate communities, such as causality and offline bandits, gives additional didactic value to the readers in its mapping of popular alignment methods DPO and PPO to counterfactual estimators, and the proposed method is sound. I think quite a few people might be interested in building on top of this work.

So there appears to be quite some potential in the paper, which has definitely been worked out seriously. On the other side, we have some criticism that has not been substantially changed by one reviewer, who is concerned with: 1) reproducibility of the experiments; 2) generalisability of the approach (validation is experimental and the experiments themselves are only partially described); 3) motivation of the need for fine-tuning.

In my view point 3 is not so critical (as pointed out by another reviewer), so we are left with 1 and 2. My viewpoint is that this should not be enough to dismiss the paper at this point; it's probably something that can be partly corrected by the authors in the revision and partly afterwards. Moreover, the authors are aware that it's in their main interest that the experiments are correct. So I regard the partial lack of thoroughness in the experiments as a result of lack of time or a bit of naivety.